# Listening and Processing Skills in Young School Children with a History of Developmental Phonological Disorder

**DOI:** 10.3390/healthcare12030359

**Published:** 2024-01-31

**Authors:** Nelli Kalnak, Cecilia Nakeva von Mentzer

**Affiliations:** 1Department of Women’s and Children’s Health, Karolinska Institutet, 171 77 Stockholm, Sweden; 2Department of Speech-Language Pathology, Helsingborg Hospital, 251 87 Helsingborg, Sweden; 3Department of Neuroscience, Uppsala University, 751 23 Uppsala, Sweden; 4School of Health Sciences, Örebro University, 701 82 Örebro, Sweden; 5SpecEDL—Special Education, Development and Learning, Örebro University, 701 82 Örebro, Sweden; 6Disability Research, Örebro University, 701 82 Örebro, Sweden

**Keywords:** children, listening and processing skills, Developmental Phonological Disorder (DPD), Evaluation of Children’s Listening and Processing Skills (ECLiPS)

## Abstract

There is a lack of longitudinal studies on the broad-based outcomes in children with Developmental Phonological Disorder (DPD). The aim of this study was to investigate listening and processing skills in a clinical sample of 7-to-10-year-old children diagnosed with DPD in their preschool years and compare these to same-aged typically developing (TD) children. The Evaluation of Children’s Listening and Processing Skills (ECLiPS) was completed by parents of 115 children with DPD and by parents of 46 TD children. The total ECLiPS mean score, and the five subscale mean scores, the proportion of children with clinically significant difficulties (≤10th percentile), and the proportion of children with co-occurrence of clinically significant difficulties on more than one subscale, were calculated. Results showed that the ECLiPS mean scores did not differ between the groups. There was no difference between groups regarding language and literacy, but a higher proportion of children with DPD than TD had difficulties in the total score, speech, and auditory processing, environmental and auditory sensitivity, and pragmatic and social skills. In addition, 33.9% of children with DPD had clinically significant difficulties in two or more subscales compared to 10.9% of TD children.

## 1. Introduction

Developmental Phonological Disorder (DPD) is characterized by difficulties in three closely interrelated aspects of the spoken sound system: perception, production, and representation of speech sounds. There is variability within each aspect of difficulties across children with DPD, with some children showing mild difficulties within a single aspect, and others showing a broader picture of difficulties within or across perception, production, and representation. DPD is diagnosed in the absence of any apparent acquired oral motor problems, and in the absence of hearing loss [1].

DPD and Speech Sound Disorders (SSD) in childhood are often used interchangeably [2,3,4]. However, SSD includes both functional and organic difficulties with the spoken sound system [5]. Functional SSD comprises articulation disorders (e.g., distortions and substitutions in the production of individual speech sounds) and phonological disorders (i.e., rule-based errors such as fronting, stopping, and deletions affecting more than one speech sound). Organic SSD includes motor speech disorders such as childhood dysarthria or childhood apraxia of speech [6], structural disorders such as cleft palate [7], and sensory disorders such as hearing loss [8]. Articulation disorders and phonological disorders thus comprise functional SSD. Since they can often be difficult to differentiate, they are often referred to as SSD by researchers. Organic SSD, on the other hand, is referred to according to the underlying cause (e.g., motor speech disorders). In the present study, the focus is on children with DPD, i.e., functional SSD. We will use the term DPD when we refer to the participants in the present study.

The estimated prevalence of SSD has been reported at 3.4% to 5.5% in 4-to 8-year-old children [2,9,10]. However, the parent-reported prevalence of SSD is higher. McLeod and Harrison [11], who collected data regarding speech and language impairment from 4983 children in The Longitudinal Study of Australian Children, found that 11.8% of the parents of 4-to-5-year-old children were concerned about how their child talked and made speech sounds. Parents who were concerned reported “speech not clear to others” as the most frequent area of difficulty. This statement captures the most common symptom of SSD—reduced speech intelligibility—that has negative consequences on a child’s social communication [12]. Lohmander et al. [13] collected normative data from 443 Swedish children’s speech production and found that at 7 years of age, 98% of the children had reached adult speech norms. In Wren et al.’s [10] prospective population study of 7390 8-year-old British children, 86.6% had typical speech. These studies show that the majority of 7-to-8-year-old children have intelligible speech, and also that there are differences in speech development between languages; also see [14]. Moreover, most studies of SSD/DPD are based on English-speaking children. Studies in other languages than English are of importance for both assessments and interventions of DPD.

Clinicians often encounter the question of whether children can outgrow speech errors without intervention from speech-language pathologists (SLPs). To et al. [15] studied prognostic factors and time to normalization in Cantonese-speaking preschoolers who were at risk for SSD. Stimulability, i.e., a child’s ability to modify speech production when stimulated by a clinician by models and cues, and severeness of reduced intelligibility, were found to be useful prognostic factors, while atypical speech errors and expressive language abilities were not. In children who did not receive intervention, the time to normalization was estimated to be 6.6 years. These authors stress that children with poor stimulability and low intelligibility should be prioritized for SLP services. Kalnak et al. [16] found that phonological processing difficulties, as tested with nonword repetition, did not resolve as well as speech production. Poor quality of phonological representations has also been reported in several previous studies [17,18,19], and the link between low quality of phonological representations, phonemic awareness, and reading disorders has been established for many years [20]. Altogether, these results show that there is heterogeneity within children with DPD, and that many of the problems related to the spoken language system are not resolved when children reach school age. Therefore, finding methods that capture these broad-based difficulties, are warranted, motivating the current study.

The ECLiPS was developed to cover the broader range of areas of difficulties often reported by parents of children with auditory processing disorder (APD), such as listening, language, literacy, social, or hearing difficulties [21]. It is well-known that the diagnostic category of APD is under question, and that different audiological societies have different inclusion and exclusion criteria [22,23]. However, one thing that researchers and organizations do agree upon is the heterogeneity of APD. Thus, within the APD category, there is variability within each aspect of difficulties across listening, language, literacy, social, or hearing difficulties; see, for example, [23]. There are very few studies that have specifically investigated children with DPD in relation to these aspects. But we have found three studies, one large-n study [24] and two small-n studies [25,26] of other clinical populations and children with APD/Listening Difficulties, that we will present in the following section.

Petley et al. [24] investigated 146 American children 6–13 years of age with normal audiograms within the SICLiD study (Sensitive Indicators of Listening Difficulties). One group had caregiver-reported listening difficulties (LiD *n* = 67), and one group age-matched typically developing (TD) children (*n* = 79). Besides the ECLiPS, Children’s Communication Checklist, CCC-2 [27], tests of auditory processing, speech hearing in noise, and cognition were assessed. In the study, scaled scores (mean 10, SD 3) for the ECLiPS were presented (for conversions from raw scores to scaled scores for British children, see Appendix 3 in [21]). Results showed that the American TD children performed significantly better than the UK norms. When comparing the LiD group with the TD group, weaker results were observed in the LiD group regarding all comparisons. As for the ECLiPS, much lower scores were observed for the total score and one subscale score, Speech and Auditory Processing, in children with LiD. The authors conclude that children with LiD had impaired performance on a broad range of auditory and cognitive tasks relative to TD children. These authors stress that impaired cognitive function had a major influence on listening difficulties in school-aged children with normal audiograms. However, it is important to note that the classification of the LiD group was based on the outcomes of the ECLiPS questionnaire [24] (p. 14).

Barry et al. [25] sought to determine whether the ECLiPS contributed anything more to the clinical assessment of APD than three other available questionnaires used by clinicians in Australia. These three were a self-report for children with hearing loss [28], a teacher-report measure for children using hearing aids or cochlear implants [29,30], and a parental report measure [31]. Thirty-five children (7–12 years) who were referred to an audiology clinic due to suspected APD were assessed with these four questionnaires and compared to TD children. Twelve children meeting the diagnostic criteria for APD based on a commonly used clinical test battery were identified. The study showed that all questionnaires were equally sensitive to the presence or absence of some form of difficulty, requiring APD referral, and scores on all questionnaires showed more symptoms in referred children relative to TD children. However, none of the questionnaires could discriminate between children meeting the diagnostic criteria for APD and those who did not. Barry et al. discuss several reasons for this lack of discrimination, some of them possibly related to that clinical tests for auditory processing lack construct validity. Also, they reason that other factors such as cognitive ability, including attention and environmental experience, are influencing the results. Barry et al. [25] stress that the children meeting the diagnostic criteria were characterized by lower non-verbal intelligence and poorer attention than the children not meeting the APD criteria. Finally, Barry et al. [25] agree with [22] that the term APD should be viewed as an umbrella term for a range of different auditory difficulties also called Listening Difficulties, see [32,33], that may occur alone or in combination with other difficulties but ultimately affect the ability to understand spoken language.

In a study by Purdy et al. [26], a comparison of the ECLiPS to the Auditory Processing Domains Questionnaire (APDQ) is summarized with respect to which of the two parent questionnaires aligned most closely with the child’s clinical diagnosis of APD, attention deficit hyperactivity disorder (ADHD), nonspecific learning disorders, and language disorder. The sample consisted of 42 children that were referred for suspected APD. The APDQ was quicker to administer and was more accurately correlated with the clinical diagnosis. However, the ECLiPS is a screening tool with the main purpose of identifying children’s listening and processing difficulties on a broad basis [21]) and not to differentiate between diagnoses. As such, using the ECLiPS may be useful in discovering areas of difficulty in children who are diagnosed with DPD.

The ECLiPS was translated to Swedish in close collaboration with Dr. Johanna Barry, who is the originator of the English version together with Prof. David R. Moore. The translation process is fully described in a Swedish master thesis by Forsberg and Ohtamaa [34]. In that study, comparisons were made between the Swedish TD group and the English reference group [21] according to the four age groups: 7, 8, 9, and 10 years and gender. High consistency was found for boys between the Swedish and English groups in all ages and in the majority of subscales. Where inconsistencies were found, the Swedish TD group showed fewer difficulties than the English reference group. For girls, high consistency was found between the two samples in all subscales for 8-year-old girls, while the consistency in the other age groups varied. Forsberg and Ohtamaa also compared the presence of ear infections and hearing/listening problems in children with DPD and TD children and found that it was more common with these problems in children with DPD. The current study is an attempt to decrease the lack of longitudinal broad-based studies in children with DPD by collecting data with the parental questionnaire ECLiPS (Evaluation of Children’s Listening and Processing Skills) [21].

### Aims of the Study

The specific aim of the present study is to explore listening and processing skills in a clinical population of 7-to-10-year-old children who were diagnosed with DPD in preschool-age and responses to that of same-age typically developing (TD) children. The research questions were: Do 7-to 10-year-old children with a history of DPD differ from same-age TD children on results from the ECLiPS questionnaire, and if so, how?

## 2. Materials and Methods

### 2.1. Participants

In the present study, the term DPD is used for children who have been clinically diagnosed by a SLP with F80.0 Phonological disorder (in Swedish: F80.0A Fonologisk språkstörning) or with F80.1 Expressive language disorder (in Swedish: F80.1B Fonologisk och grammatisk språkstörning). The latter is used when a phonological disorder co-occurs with grammatical difficulties (The National Board of Social Affairs and Health, https://klassifikationer.socialstyrelsen.se accessed on 19 January 2023).

Participants with DPD were recruited from two hospital SLP clinics in central Sweden; see Figure 1. One SLP clinic belongs to a hospital in a larger city (≈500,000 inhabitants), and the other SLP clinic belongs to a hospital in a smaller city (≈200,000 inhabitants). Both hospitals covered urban and rural areas. Participants were identified by access to visiting statistics. The participants were between 7:0 and 10:11 years of age at time of the recruitment and participation in the present study. They were diagnosed with DPD between 3 to 6 years of age. In the larger city hospital clinic, all first 3–4 children born monthly in the years 2006 to 2009 (*n* = 123) with DPD were invited. In the medium city hospital clinic, all children born in the years 2006 to 2009 (*n* = 667) with DPD were invited. Information about the study, a letter of informed consent, and the ECLiPS questionnaire were sent to the caregivers of 790 children. In total, 115 families responded with written consent and a completed ECLiPS questionnaire, corresponding to a response rate of 14.6%. See Figure 1 for flow chart of the recruitment procedure of children with DPD.

The TD children, who formed the comparison group, were recruited from four mainstream schools in a medium-sized city, also in central Sweden. The inclusion criteria were: age 7 to 10 years, typical speech and language development, normal hearing, and not having received or was receiving any special education according to parents and/or teachers. Information about the study, a letter of informed consent, and the ECLiPS questionnaire were sent to the caregivers of approximately 110 children. Forty-eight consents and completed ECLiPS questionnaires were received, corresponding to a response rate of 43.7%. However, two of these were excluded because one child did not fulfill the lower age criteria, and one child’s parents had noted “some problems with pronunciation and forming sounds”. Altogether, 46 participants formed the group of TD children.

The mean age in the total sample of the DPD group was 9 years (min–max = 7.0–10.5 years, months), while the TD group had a mean age of 8.5 years (min–max = 7.0–10.6 years, months). An independent-sample *t*-test showed that the difference in age between groups was significant (*t*(159) = 3.3, *p* = 0.003, two-tailed). There was no statistically significant difference between groups for the proportion of female and male participants, with the DPD group having 68.7% male participants as compared to 60.1% in the TD group (χ^2^ (1) = 0.78, *p* = 0.342); see Table 1. Of the 115 children with DPD, 81.7% (*n* = 94) were diagnosed with F80.0A and 18.3% (*n* = 21) with F80.1B. There was no statistically significant difference between children with F80.0A or F80.1B regarding their mean age (*t*(113) = −0.52, *p* = 0.60), nor gender (χ^2^ (1), = 0.089, *p* = 0.77).

### 2.2. Procedure

The ECLiPS consists of 38 statements covering five subscales: (1) speech and auditory processing, SAP (9 items), e.g.,: *“When there is a sudden noise, is confused about where to look”,* and “*Seems deaf when lots of people are talking*”, (2) Environmental and auditory sensitivity, EAS (8 items, e.g.,: *“Complains about loud sounds”*, and *“Becomes upset if daily routine is changed”*, (3) Language, literacy, and laterality, LLL (6 items), e.g.,: *“Writes numbers down wrong”*, *“Says some words wrong”*, (4) Pragmatic and social skills, PSS (6 items), e.g.,: *“Has obsessive interests”* and *“Says some sentences, or words, over and over”* and (5) Memory and attention, MA (8 items), e.g.,: *“Finds it difficult to do more than one thing at a time”* and “*Needs things repeated to understand”.* All 38 statements were analyzed in the present study, and we did not exclude any questionnaires (see [21] for validation procedure).

The parents are instructed to respond to how well each statement corresponds to their child’s behavior by a five-point Likert scale: strongly agree (2), agree (1), neither or (0), disagree (−1), and strongly disagree (−2). The responses *strongly agree* and *agree* indicate more difficulties. Cronbach’s alpha coefficient values for the subscales vary between 0.83 and 0.94, see Table 4.8 in [21] suggesting that the items within the subscales have an acceptable level of internal consistency. The ECliPS questionnaire also includes a short medical survey, and one of the required items is a commentary field regarding if the child had any developmental diagnoses or concerns. This item was used for the inclusion criteria for the TD group.

### 2.3. Data Analysis

The data analyses were performed in the Statistical Package for the Social Sciences (SPSS, Version 28). The data were analyzed using descriptive and inferential statistics. Mean, median, and standard deviation (SD) for the total raw scores and subscale raw scores were obtained for each group. The Mann–Whitney U-test was used for comparisons where group sizes were unequal, and the data were on the ordinal level. Otherwise, independent sample t-tests were used. *p*-values of <0.05 were considered significant. Following the ECLiPS manual [21] (chapter 2), a cut-off value at or below the 10th percentile i.e., an age-scaled score of around 6, was used for categorization of results defined as clinically significant difficulties. Results of the children with the diagnosis code F80.0A (*n* = 94) were compared to children with F81.0 (*n* = 21) for all measures. There was no significant difference between these groups (*p*-values range = 0.31 to 0.70), and scores from the two diagnostic groups were, therefore, collapsed in all analyses.

## 3. Results

### 3.1. Total ECLiPS Scores and Subscale Scores

No significant differences were found between the DPD and TD groups in any of the comparisons; see Table 2.

Associations with age were investigated. In the DPD group, Pearson’s correlation analysis showed only one significant correlation with weak strength, namely between age and the LLL subscale (*r* = −0.25, *p* = 0.008); see Figure 2. Three of these children (two 7-year-olds and one 8-year-old, case numbers 65, 83, and 96) showed tendencies to be outliers as they were the only children who had scores > 1 (=agree). These scores were therefore removed from the analysis. This resulted in a weaker but still significant correlation with age (*r* = −0.19, *p* = 0.04). Thus, younger children were reported with higher LLL scores. Corresponding analysis for the TD group showed no significant associations with age.

Sex comparisons were made and showed more difficulties in boys than in girls within both groups. The magnitudes of all differences between boys and girls were large. For children in the DPD group, there was a statistically significant difference between boys and girls for two subscale scores. Boys were reported with more difficulties than girls in PSS and MA: PSS (boys: *M* = −0.85, *SD* = 0.91; girls: *M* = −1.28, *SD* = 0.63) *t*(113) = 2.5, *p* = 0.012 two-tailed, Cohen’s *d* = 0.84) and MA (boys: M = −0.73, *SD* = 0.90; girls: *M* = −1.10, *SD* = 0.79) *t*(113.0) = 2.1, *p* = 0.04 two-tailed, Cohen’s *d* = 0.87). For the TD group, there was a statistically significant difference between boys and girls for one subscale score: MA (boys: *M* = −0.80, *SD* = 0.66; girls: *M* = −1.23, *SD* = 0.62) *t*(44) = 2.2, *p* = 0.03 two-tailed, Cohen’s *d* = 0.64). Thus, boys were reported with more difficulties.

### 3.2. Proportions of Participants with Clinically Significant Difficulties (≤10th Percentile)

Children with DPD more often performed at or below the cut-off of the 10th percentile for the total raw score (31.3%) and for the subscales SAP (30.0%), EAS (21.7%) and PSS (16.5%), as compared to TD children (10.9%, 10.9%, 8.7%, 4.2%). The odds for children with DPD to be reported with difficulties were 2.6 to 3.9 times higher than for TD children. The highest odds were found for the subscale PSS (3.9) and the total score (3). Significant differences were observed for all comparisons except LLL and MA. There was no difference between DPD and TD in the proportions of children with results below the cut-off for the subscales LLL and MA; see Table 3.

The number of participants with zero, one, two, three, or more subscale scores with clinically significant difficulties (scores ≤ 10th percentile) and odds ratios for all number of affected subscales are presented in Table 4.

First, it should be noted that over half of the children in both groups did not have any subscale scores below the cut-off (0 subscales: DPD, 51.3%; TD, 56.5%; Chi^2^ = 0.36, *p* = 0.55). In the DPD group, the children with subscale results below the cut-off were distributed across all numbers of affected scale scores. On the other hand, when children in the TD group had results below the cut-off, they mostly had difficulties in only one subscale (*n* = 16, 34.8%).

This means that in the DPD group, it was more common to find children with more than one affected subscale compared to the TD group. In addition, the DPD group had a higher proportion three times that of children with two or more subscales below the cut-off (33.9%) as compared to the TD group (10.9%). This result shows that it was three times as common to find a broad variety of listening and processing difficulties, as assessed by the ECLiPS, in children with DPD. More specifically, it was three times more common for children with DPD to have two subscales below the cutoff, and four times more common to have all five subscales below the cutoff in the DPD group as compared to the TD group. However, it should be noted that significant differences were observed only for one subscale—more often reported in the TD group—and for two or more subscales, more often reported in the DPD group.

### 3.3. Co-Occurrence of Subscales with Clinically Significant Difficulties

Among children with one affected subscale, LLL was the most common subscale in both groups (DPD, *n* = 10; TD, *n* = 6), followed by MA (DPD, *n* = 3; TD, *n* = 4). DPD participants with two subscale scores at or below the 10th percentile usually involved MA and any one of the other subscales (*n* = 5). When a co-occurrence of three subscales was found, the most common combination was MA with any of the other subscales (*n* = 4).

## 4. Discussion

The present study is the first study reporting listening and processing skills in young school-age children who were diagnosed with DPD in pre-school age. It is also the first study using the ECLiPS with an overall large sample size as compared to earlier studies of clinical populations with APD/LiD [24,25,26]. It is also the first study using ECLiPS in a non-English sample.

The overall purpose was to add to the understanding of the prognosis in children with DPD. The research questions were: Do 7-to-10-year-old children with a history of DPD differ from same-age TD children on results from the ECLiPS questionnaire, and if so, how?

### 4.1. Core Results: Differences with Respect to Overall Mean Scores

The answer to our research question regarding if there are any differences between groups is both yes and no. Children with DPD do not differ from children with TD when comparing the mean raw scores. This means that we don’t see overall difficulties with listening and processing skills in children with DPD in the present study, but the large variation within the group suggests that subgroups may well do. These results do not corroborate with the findings in the SICLid study [24]. In that study, overall, more difficulties were observed in children with LiD on the total score and on speech and the auditory processing of the ECLiPS compared to children with TD. Also, lower scores were observed on the CCC-2. However, in both the present study and [24], the TD groups had overall higher scores than those reported for the UK norms [21]. In both Petley et al.’s. study [24] and the present study, the children in the respective study’s TD group were recruited based on caregiver-reported typical development. One important factor distinguishing our study from [24] is that they included assessments confirming that their TD group had cognitive abilities within the normal range based on a standardized battery of tests. Results from their background questionnaire reporting “Other cognitive concerns” (number of individuals who had one or more caregiver-reported cognitive diagnoses, interactions, or reports from special needs professionals) in their study show that three of the TD children (3.8%) had received some form of speech or language interventions at school, which was not reported for the TD children in our study.

### 4.2. Associations with Age and Gender Comparisons

A significant association between age and LLL was observed only in the DPD group, not the TD group. Thus, more difficulties were observed in younger children with DPD. This association can be interpreted as that language and literacy need more time to mature in children with DPD; see [35,36] for similar results.

Sex comparisons showed more difficulties in boys regarding memory and attention in both groups and with a large effect size. However, there was only a difference between boys and girls for pragmatic and social skills in the DPD group. Again, boys were reported to have more difficulties, and, again, the effect size was large. It is well-established that in clinical samples, the sex ratio of those who have a profile of speech and language disorders is higher for males [37,38,39].

### 4.3. Differences with Respect to Clinically Significant Difficulties

When investigating the proportions of participants with poor results, the findings show that DPD participants were more often reported having sub-threshold scores for the total score, and the subscale SAP, EAS, and PSS scores. The finding that the subscale environmental and auditory sensitivity differed between the groups, to the disadvantage of the children with DPD, corroborates the findings of hearing-related problems in DPD children that were found by [3]. In that study, subclinical hearing loss was found in 4–5-year-old children with DPD. The odds for children in the DPD group to have scores at the sub-threshold of the ≤10th percentile for the total ECLiPS score, and for the EAS, SAP, and PSS subscale scores, were 2.6 to 3.9 times higher than for the TD group, and all of these results were statistically significant. To conclude, the results show a less positive prognosis for children with DPD, since they indicate an increased risk of a wide range of difficulties related to listening and processing skills. There was no difference in the proportion of children in the DPD and TD groups with poor results in neither the LLL nor the MA scales.

### 4.4. The Results Regarding Type of Subscales Involved When Below the Cut-Off ≤ 10th Percentile

In both groups, LLL stands out as the subscale with the highest percentage of children with results below the cut-off (DPD 36.5%; TD 30.4%). Such a high amount of TD children who have clinically significant problems according to the cut-off in the ECLiPS manual indicates a low specificity of the ECLiPS as a screening tool in a Swedish context. The TD participants were not formally assessed, so it cannot be ruled out that some of them did not have typical speech and language development, which then would explain their poor results on the LLL scale. But the TD group’s results on LLL might as well reflect that there is a large normal variation of the items included in the LLL subscale, rather than clinically significant difficulties. Also, if all children in the DPD group had a current DPD status, an even higher proportion would have been expected to have results below the cut-off on the LLL subscale. They do not, which indicates that most children (about 65%) have recovered from their DPD. Moreover, about a third of the parents report clinically significant difficulties in listening and processing skills for the total score, as well as for the SAP subscale score. SAP covers a broad range of abilities, e.g., sound localization, speech-processing speed, and phonological processing. This confirms that about a third of the children in the DPD group still have DPD. Additionally, since the cut-off in the ECLiPS manual, a score below the 10th percentile derives from being generally considered as clinically significant, and does not result from empirical analysis, these findings need to be interpreted with caution.

### 4.5. Limitations

One limitation of the present study is the sample sizes, especially of the TD group. Another is the lack of assessments to confirm that the children in the TD group do not have, for example, speech or language disorders. Also, since the DPD group was not formally assessed at the time of participation in the study, we do not know if they still have DPD. We did not gather information about the type or degree of difficulty of DPD. Hence, the results reflected that of the broader group of children with DPD, the clinical picture can change, new clinical diagnosis can be added, and diagnosis can be removed due to catch-up after intervention, for example.

### 4.6. Future Implications

In future research, if the implementation of the Swedish version of ECLiPS is realized, an item-level analysis within the LLL scale, for example, could be performed. This would then provide an understanding of the type of LLL items that caused about a third of children in the DPD and TD groups to have difficulties below the cut-off. Item-level analysis could also investigate possible differences in the type of LLL difficulties between the DPD and TD groups and, most importantly, contribute to an understanding of whether the results have any clinical relevance.

## 5. Conclusions

The present study shows that this clinical sample of 115 school children who were diagnosed with DPD in their preschool years showed higher odds of having listening and processing difficulties as compared to 46 TD children. The findings indicate that DPD might have long-term consequences for a broad picture of difficulties related to speech and auditory processing, pragmatic and social skills, and environmental and auditory sensitivity. This must be addressed in a longitudinal design.

The purpose of the ECLiPS is to be used as a screening of a broad set of listening and processing difficulties, and if a child shows results below the cut-off, they should be considered for further clinical assessments. However, since almost half of the children in the TD group were reported with difficulties at or below the cut-off in at least one subscale, the findings could indicate that the ECLiPS might lack the clinical accuracy of a screening tool in a Swedish context. The present TD group’s results on the ECLiPS need to be replicated before any generalization of the findings can be stated.

## Figures and Tables

**Figure 1 healthcare-12-00359-f001:**
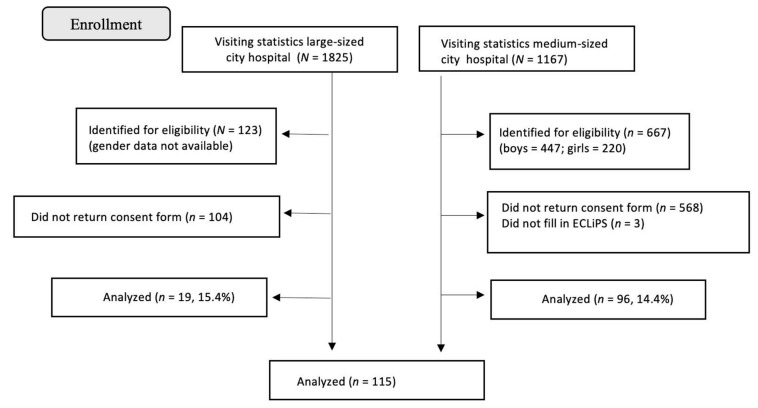
Flow chart of the recruitment process at the SLP clinics.

**Figure 2 healthcare-12-00359-f002:**
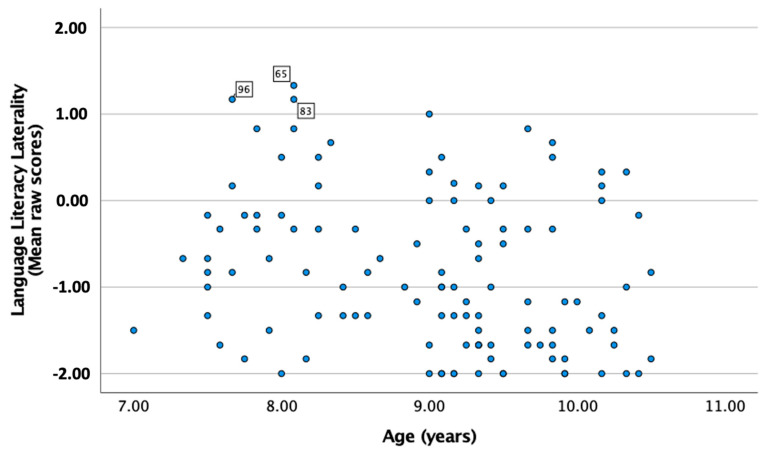
Scatterplot with age (*x*-axis) and the LLL subscale (*y*-axis) in the DPD group.

**Table 1 healthcare-12-00359-t001:** Age and gender ratio in children with DPD and TD children.

	DPD (*n* = 115)	TD (*n* = 46)	*p*-Value
Age mean, years, months (range)	9.0 (7.0–10.5)	8.5 (7.0–10.6)	0.003
Female-to-male ratio (%)	36:79 (31.3:68.7)	18:28 (39.1:60.9)	0.342

**Table 2 healthcare-12-00359-t002:** Mean, median, SD, and Min–Max raw scores for children in the DPD group (*n* = 115) and children in the TD group (*n* = 46).

	DPD				TD				
	Mean	Median	SD	Min–Max	Mean	Median	SD	Min–Max	*p*-Value
Total raw score	−1.01	−1.16	0.77	−2.00–1.46	−1.22	−1.26	0.50	−1.97–0.30	0.24
SAP	−1.17	−1.33	0.84	−2.00–1.56	−1.44	−1.62	0.53	−2.00–0.22	0.19
EAS	−1.17	−1.63	0.98	−2.00–1.75	−1.33	−1.63	0.80	−2.00–1.13	0.68
LLL	−0.84	−1.00	0.9	−2.00–1.33	−0.99	−1.17	0.83	−2.00–0.67	0.33
PSS	−0.98	−1.00	0.85	−2.00–2.00	−1.25	−1.33	0.63	−2.00–0.17	0.09
MA	−0.85	−0.88	0.88	−2.00–1.13	−0.97	−1.00	0.67	−2.00–0.50	0.60

Note, SAP = Speech and Auditory Processing. EAS = Environmental and Auditory Sensitivity. LLL = Language/Literacy/Laterality. PSS = Pragmatic and Social Skills. MA = Memory and Attention.

**Table 3 healthcare-12-00359-t003:** Percentage (number) of participants with a total score and/or subscales scores at or below the 10th percentile.

	DPD % (*n*)	TD % (*n*)	χ^2^	*p*-Value	Odds Ratio
Total score	31.3 (36)	10.9 (5)	6.5	0.01	3.0
SAP	30.0 (31)	10.9 (5)	4.2	0.04	2.8
EAS	21.7 (25)	8.7 (4)	*	0.03	2.6
LLL	36.5 (42)	30.4 (14)	0.4	0.53	1.2
PSS	16.5 (19)	4.3 (2)	*	0.03	3.9
MA	22.6 (26)	13.0 (6)	1.5	0.22	1.8

Note, SAP = Speech and Auditory Processing. EAS = Environmental and Auditory Sensitivity. LLL = Language/Literacy/Laterality. PSS = Pragmatic and Social Skills. MA = Memory and Attention, * = Fisher exact probability test, one-tailed. Please note that the same children may appear in more than one of the subscales.

**Table 4 healthcare-12-00359-t004:** Percentage (number) of participants with clinically significant difficulties (≤10th percentile) in none, one, or more of the ECLiPS subscales in the DPD and TD groups, respectively.

No of Subscales	DPD % (*n*)	TD % (*n*)	χ^2^	*p*-Value	Odds Ratio
0 subscales	51.3 (59)	56.5 (26)	0.36	0.55	0.9
1 subscale	14.8 (17)	34.8 (16)	8.07	<0.01	0.4
2 subscales	13.0 (15)	4.3 (2)	*	0.15	3.0
3 subscales	8.7 (10)	4.3 (2)	*	0.51	2.0
4 subscales	3.5 (4)	0.0 (0)	*	0.32	3.5
5 subscales	8.7 (10)	2.2 (1)	*	ns	4.0
Sum of 2 or more	33.9 (39)	10.9 (5)	8.8	<0.01	3.1

* = Fisher exact probability test, one-tailed. ns = non significant.

## Data Availability

Data may be demonstrated on request from the second author.

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
