# Peer review of "Listening and Processing Skills in Young School Children with a History of Developmental Phonological Disorder"

_healthcare, 2024, doi:10.3390/healthcare12030359_

Round 1

Reviewer 1 Report

Comments and Suggestions for Authors

Line 196: It is stated that ‘We did not gather information about the type or degree of difficulty of DPD’ ‘relation to claimed sub-types. Hence, the results reflected that of the broader group of children with DPD irrespective of the different categories of DPD as claimed by some researchers (e.g., Dodd (2005), or the severity of the disorder.

Lines 366-367: It is stated that the TD group was not assessed for speech and language disorders. Yet in lines 203-204 it is stated that one of the inclusion criteria was, ‘(ii) typical speech and language development’ …this inconsistency needs to be clarified. Who reported/assessed their speech and language skills?

Also, who reported or assessed their hearing acuity? Was a history of fluctuating hearing loss eliminated? It is well documented in the literature that children with a history of otitis media with effusion with the possibility of a mild fluctuating hearing loss are associated with having auditory processing difficulties?

Although there is a section (5) dedicated to limitations, yet some of these are mentioned elsewhere in the text.  These may be better grouped under one section.

Reviewer 2 Report

Comments and Suggestions for Authors

The authors have used the ECLiPS questionnaire in children with DPD to understand their listening and processing skills. The authors have used the Swedish version of the questionnaire. These types of research are required. I have several concerns with this manuscript.  

There is a lot of literature on DPD which may not be relevant to this study. Why do the authors want to test children with DPD only? What do the authors want to capture using the ECLiPS questionnaire in children with DPD? Why do the authors think that ECLiPS is a far better questionnaire than APDQ or other questionnaires?

There is a large difference in the sample size between groups. I do not see the proper justification for such a huge difference in the sample size. The authors did not control important variables such as speech-language and hearing in both groups.

Lines 42: The year is missing for the ASHA reference.

Line 134-136-: This sentence is not clear. Incorporate noise tests?

Lines 174-176: I do not see specific answers for this question in the discussion. The ECLiPS questionnaire was given multiple times to fill out or just one time?

Lines 203-205: How did the authors know that TD children’s speech, language, and hearing are normal?

Lines 214-215: If there are 172 (TD= 47 and DPD = 115) participants then the df should be 160 instead of 157.

Data analysis:

The 10th percentile cut-off value was taken from Barry and Moore (2015). Or from the Swedish norm?

The magnitude of the difference between the two groups is small. Can this be simply due to differences in sample size?

Table 3: This table raises a lot of questions about the ECLiPS questionnaire. In the TD children group, in three out of five sub-scales, 10% of children’s performance is significantly lower. 31 out of 47 TD children’s score is significantly lower. I am not sure how many of these children have failed in all five subscales or if it is different children. Did Barry et al (2015) report similar findings in TD children? Or Forsberg and Ohtamaa (2019)?   

Lines 269-271: Very weak correlation. The authors should plot this graph. Did the authors control extreme values? Extreme values will influence correlation.

From this study, I cannot understand what is the take-home message. Should someone use the Swedish version of the ECLiPS questionnaire to understand listening and processing skills in children suspected of speech and language difficulties?

Why can’t the authors examine the listed future implications from the existing data?

The conclusion of the study does not match the results. ~ 30% of TD children have significant difficulty in the LLL subscale. This must be reflected in the abstract as well.

Comments on the Quality of English Language

Requires very minor edits.

Round 2

Reviewer 1 Report

Comments and Suggestions for Authors

I would suggest the following to be included as a LIMITATION:

Information about the type or degree of difficulty of DPD was not gathered. Hence, the results reflected that of the broader group of children with DPD which could be stratified given that there may be children with a particular sub-typo of DPD who could have more difficulties than those having other sub-type/s. 

Author Response

Please see attached file for our answers to R1 and R2.

Reviewer 2 Report

Comments and Suggestions for Authors

Some minor suggestions. 

The last paragraph before the aims of the study needs to be connected or linked to the Aims of the study. 

Lines 161-163: This sentence needs to be modified. The word prognosis is misleading here. Because the authors are not investigating the prognosis in this study. 

Comments on the Quality of English Language

The article needs some minor English editing. 

Author Response

(The authors gave the same response as above.)
